# Melatonin Inhibits Hypoxia-Induced Alzheimer’s Disease Pathogenesis by Regulating the Amyloidogenic Pathway in Human Neuroblastoma Cells

**DOI:** 10.3390/ijms25105225

**Published:** 2024-05-10

**Authors:** Nongnuch Singrang, Chutikorn Nopparat, Jiraporn Panmanee, Piyarat Govitrapong

**Affiliations:** 1Chulabhorn Graduate Institute, Bangkok 10210, Thailand; nuchsingh@gmail.com; 2Innovative Learning Center, Srinakharinwirot University, Bangkok 10110, Thailand; m_chukorn@hotmail.com; 3Research Center for Neuroscience, Institute of Molecular Biosciences, Mahidol University, Nakhon Pathom 73170, Thailand; jirapornpanmanee@gmail.com

**Keywords:** Alzheimer’s disease, beta amyloid, ischemic stroke, oxygen–glucose deprivation and reoxygenation, melatonin, hypoxia-inducible factor-1α

## Abstract

Stroke and Alzheimer’s disease (AD) are prevalent age-related diseases; however, the relationship between these two diseases remains unclear. In this study, we aimed to investigate the ability of melatonin, a hormone produced by the pineal gland, to alleviate the effects of ischemic stroke leading to AD by observing the pathogenesis of AD hallmarks. We utilized SH-SY5Y cells under the conditions of oxygen–glucose deprivation (OGD) and oxygen-glucose deprivation and reoxygenation (OGD/R) to establish ischemic stroke conditions. We detected that hypoxia-inducible factor-1α (HIF-1α), an indicator of ischemic stroke, was highly upregulated at both the protein and mRNA levels under OGD conditions. Melatonin significantly downregulated both HIF-1α mRNA and protein expression under OGD/R conditions. We detected the upregulation of β-site APP-cleaving enzyme 1 (BACE1) mRNA and protein expression under both OGD and OGD/R conditions, while 10 µM of melatonin attenuated these effects and inhibited beta amyloid (Aβ) production. Furthermore, we demonstrated that OGD/R conditions were able to activate the BACE1 promoter, while melatonin inhibited this effect. The present results indicate that melatonin has a significant impact on preventing the aberrant development of ischemic stroke, which can lead to the development of AD, providing new insight into the prevention of AD and potential stroke treatments.

## 1. Introduction

Stroke and Alzheimer’s disease (AD) are prevalent age-related illnesses; however, the connection between the two conditions remains poorly understood. Brain ischemia is the third most common degenerative disease of the central nervous system and is a leading cause of death in both developed and developing countries [1]. From 1990 to 2019, there was a 70% increase in the annual incidence of stroke and stroke-related mortality [2]. Stroke is categorized into intracerebral hemorrhage and ischemic stroke (IS), both caused by the interruption of crucial blood flow to the brain. IS occurs due to thrombotic obstruction within the brain blood vessels, which interrupts blood flow in the affected area, preventing tissues from receiving sufficient oxygen and nutrients [3]. The interruption of blood flow leads to severe consequences at the cellular level, including neuronal apoptosis and necrotic cell death [4].

AD is a progressive neurodegenerative disease characterized as the most common form of dementia that significantly affects both physical and mental health and ultimately results in death [5]. The pathological hallmarks of AD include the degeneration of cholinergic neurons, the accumulation of amyloid β (Aβ) deposits in senile plaques, and the presence of neurofibrillary tangles in the brain [6]. In vivo and in vitro studies have shown that Aβ accumulation impairs spatial learning and memory [7] and increases the levels of free radicals upon solubilization, which could result in lipid peroxidation, protein and nucleic acid oxidation, and mitochondrial dysfunction [8,9].

Both IS and AD share common risk factors and vascular and neuroinflammatory pathologies and can lead to dementia. Epidemiological studies have shown that IS increases the risk of developing AD, and conversely, AD increases the risk of developing IS [10,11,12]. There is increasing evidence suggesting that cerebral ischemia may play a role in the pathogenesis of AD [13,14]. Strokes have been associated with increased amyloid accumulation, possibly due to reduced Aβ clearance [15]. Chronic hypoperfusion affected Aβ clearance in a mouse model of AD [16]. Recently, a number of studies have suggested that an insufficient oxygen and nutrient supply in the brain during hypoxia may induce hypoxia-sensitive pathways that activate amyloid precursor protein (APP) processing [17,18]. Consequently, both IS and AD cause the accumulation of cerebral Aβ, a neurodegenerative pathology leading to dementia. Intracellular Aβ accumulation causes cell death by promoting proinflammatory responses, impairing glucose metabolism, and damaging mitochondria [19]. This suggests that there may be common pathological mechanisms involving Aβ underlying both conditions.

Ischemic conditions can be induced by depriving the tissue of oxygen and glucose or by restricting chemical and enzymatic metabolism [20]. The oxygen–glucose deprivation and reoxygenation (OGD/R) model is commonly used in studies related to ischemia and hypoxia. It induces neuronal injury through mechanisms associated with hypoxia-inducible factor 1 (HIF-1) [21]. HIF-1, a basic helix–loop–helix heterodimeric transcription factor, plays an essential role in the cellular response to hypoxia [22]. HIF-1 is also a master regulator with which cells adapt to changes in oxygen and glucose concentrations [22]. It is composed of an oxygen-regulated unit (HIF-1α) and a stable constitutively expressed subunit (HIF-1β) [23]. Hypoxia has been reported to enhance HIF-1 transcriptional activity and HIF-1α levels [23,24].

The amyloid cascade hypothesis is a preponderant hypothesis regarding the pathogenesis of AD. It posits that Aβ initiates a series of consequences that culminate in the initiation of AD. The amyloidogenic pathway indicates the series of events that create the biogenesis of Aβ. Initially, APP is cleaved by β-secretase, resulting in the production of soluble β-APP fragments (sAPPβ) and C-terminal β fragments (CTFβ, C99). Subsequently, C99 is cleaved by γ-secretase, leading to the generation of APP intracellular domain (AICD) and Aβ. Aβ exists in two fundamental forms: soluble Aβ40 and insoluble Aβ42. Β-secretases comprise beta-site APP-cleaving enzyme 1 (BACE1) and beta-site APP-cleaving enzyme 2 (BACE2), whereas γ-secretase is a complex comprising Nicastrin, anterior pharynx defective 1 (APH-1), and presenilins (PS1 or PS2). Therefore, interfering with these secretases can cause a disturbance in the synthesis of Aβ. The BACE1 gene promoter contains a biologically active region that interacts with HIF-1α, and this interaction has the potential to modulate the quantities of BACE1 protein and mRNA [25,26,27,28]. The interaction between HIF-1α and BACE1 activation at the DNA transcription level could potentially play a role in the development of AD in individuals with IS. Therefore, exploring therapeutic strategies that focus on the amyloid cascade has the potential to develop effective drugs for the treatment of AD in IS.

Melatonin, also known as N-acetyl-5-methoxytryptamine, is a hormone primarily produced by the pineal gland and released by the suprachiasmatic nucleus that regulates the circadian rhythm. Additionally, it has been recently revealed that melatonin is produced within the mitochondria of all cells [29]. Melatonin has a variety of functions, including immune modulation, antioxidant activity, and neuroprotection. Melatonin has been shown to play a key role in protecting cell membrane permeability [30] and promoting cell survival under stressful conditions, such as oxidative stress and the damage caused by free radicals. Previous studies have shown that melatonin inhibits Aβ-induced oxidative stress and cell death in neuronal cells [31,32,33]. Melatonin, a versatile compound, has exhibited neuroprotective effects in both in vitro and in vivo cerebral ischemia/reperfusion (I/R) models [34]. Our previous study showed that melatonin could reduce BACE1 expression in SH-SY5Y cells under normoxic conditions [35]. Recently, we also demonstrated that melatonin could protect against hippocampal damage and memory function impairment in rats with chronic cerebral hypoperfusion by reducing the levels of AD markers, such as Aβ and pTau [36]. This abundance of compelling evidence supports the notion that melatonin possesses the potential to serve as a therapeutic agent in the treatment of AD pathogenesis and IS.

The purpose of the present study is to investigate the amyloidogenic pathway in IS by utilizing the OGD/R model to assess the relationship between hypoxia and amyloidogenic processing, as well as to evaluate the effect of melatonin on this process.

## 2. Results

### 2.1. Effect of Melatonin on the Viability of SH-SY5Y Cells under OGD/R Conditions

To assess whether oxygen–glucose deprivation (OGD) and reoxygenation (OGD/R) had a physiological impact on human neuroblastoma cells (SH-SY5Y), we first evaluated the cell morphology by using a light microscope and cell viability by using a Muse cell viability kit and trypan blue assays. We detected a dramatic decrease in cell density after OGD but no significant changes in cell density after OGD/R compared with that of the control untreated cells (Figure 1a). The SH-SY5Y cells under OGD conditions exhibited greater isolation than those under normoxia conditions and were more spindle-shaped. Moreover, we found significantly reduced cell viability in the OGD group compared with that in the control group, while OGD/R with 50 and 100 µM of melatonin significantly (*p* < 0.05) increased the cell viability compared with that in the OGD/R group (Figure 1b,c). These results suggested that during oxygen–glucose deprivation, growth and cell viability [37,38] were reduced, whereas during reoxygenation with melatonin, these effects were reversed. This indicated that melatonin improved cell survival under OGD/R conditions.

### 2.2. Melatonin Affects the Nuclear Translocation and mRNA Transcription of HIF-1α in SH-SY5Y Cells under OGD/R Conditions

To investigate the role of ischemia in neuronal cells, the presence of HIF-1α was detected and confirmed using qPCR analysis (Figure 2a). HIF-1α mRNA was highly upregulated by both OGD and OGD/R. HIF-1α protein expression was highly upregulated under OGD conditions but was very low under OGD/R conditions (Figure 2b,c). There were two HIF-1α bands in the OGD group; however, the top band disappeared while the lower band showed less intensity under the OGD/R conditions. Previous findings from Toffoli’s group [39] reported that HIF-1α exhibited a progressive increase in its phosphorylated forms throughout the course of the hypoxic period. Two bands appeared on the Western blot, with the slower migrating line (top band) being a phosphorylated form of HIF-1α.

To investigate the effect of melatonin on HIF-1α expression in OGD/R, the cells were subjected to OGD for 4 h and then reoxygenation for 24 h with or without various concentrations of melatonin. Compared with those in the OGD/R group without melatonin, the levels of HIF-1α mRNA were significantly (*p* < 0.01 and *p* < 0.001, respectively) lower in the 10 and 50 µM melatonin groups (Figure 2a). In addition, the protein expression of HIF-1α was significantly (*p* < 0.05) decreased in the presence of 100 µM of melatonin compared with that in the OGD/R group without melatonin. Nuclear localization is a measure of HIF-1α isoform activation [40] therefore, we used immunostaining to visualize and detect HIF-1α nuclear localization. As expected (Figure 2c,d), HIF-1α was present in the nucleus of the neuronal cells under OGD and OGD/R conditions (70% and 60%, respectively) (Figure 2e), while under OGD/R with melatonin, a significant decrease in HIF-1α was found in the nucleus (60%) (Figure 2e). The present results demonstrated that melatonin reduced HIF-1α mRNA and HIF-1α protein expression under ODG/R conditions. In addition, we detected the nuclear translocation of HIF-1α under OGD/R conditions.

### 2.3. Effect of Melatonin on the Transcription Level of Amyloidogenic Processing Enzymes in the SH-SY5Y Cells under OGD/R Conditions

During hypoxia, HIF-1α is stabilized and translocated to the nucleus, where it activates many genes involved in apoptosis [41] and the amyloid pathway [26]. To answer the question of whether ischemia is linked to AD, we determined the mRNA levels of different APP-processing enzymes under OGD and OGD/R conditions with or without melatonin. Our results showed that OGD/R conditions increased the mRNA levels of BACE1, presenilin-1 (PS1), and APP but not of TAU, whereas 50 µM of melatonin significantly (*p* < 0.05) decreased the mRNA levels of the APP-processing enzymes (BACE1 and PS1) (Figure 3a–c), and 10 and 50 µM of melatonin significantly increased the tau mRNA level compared with that under OGD/R conditions without melatonin (Figure 3d). We next considered whether OGD/R could affect BACE1 expression at the transcriptional level by regulating the activity of the BACE promoter. A luciferase assay was performed to determine the effect of BACE1 promoter activation. The SH-SY5Y cells were transfected with the BACE1 promoter–luciferase reporter construct (pGL3-BACE1-luc). The transfected cells were then exposed to OGD/R with or without 100 μM of melatonin for 24 h. The results of our study showed that the BACE1 promoter activity significantly (*p* < 0.05) increased under OGD/R conditions, while treatment with 100 µM of melatonin diminished the increased BACE1 promoter activity caused by OGD/R (Figure 4a,b). Our results indicated that hypoxia regulated BACE1 transcription, whereas melatonin decreased hypoxia conditions and the presence of APP-cleaving enzyme genes.

### 2.4. Effect of Melatonin on the Amyloid Pathway and Tau Protein in SH-SY5Y Cells under OGD/R Conditions

APP is sequentially cleaved first by β-secretases (BACE1), yielding C99, and then by γ-secretases (PS1), yielding Aβ peptides. Hypoxia upregulated the mRNA levels of the major amyloidogenic processing enzymes. Then, the protein expressions were also determined. Our result (Figure 5a–c) showed that hypoxia upregulated BACE1 and PS1 protein expression. OGD/R significantly (*p* < 0.05) decreased BACE1 (Figure 5a,b) and PS1 (Figure 5a,c). OGD/R with 50 and 100 of µM of melatonin significantly (*p* < 0.05) decreased BACE1 (Figure 5b), while 10 and 50 µM of melatonin significantly (*p* < 0.05) decreased PS1 compared to OGD/R without melatonin.

The reasons why we looked at the pTau levels have been explicated. AD is characterized by the presence of extracellular beta-amyloid (Aβ) plaques and intracellular neurofibrillary tangles (NFTs), both of which are caused by the accumulation of misfolded Aβ peptides and abnormally hyperphosphorylated tau, respectively. Our result (Figure 5d–f) indicated that hypoxia significantly upregulated pTau. OGD/R with 50 and 100 µM of melatonin significantly decreased pTau expression (Figure 5d), and 100 µM significantly decreased pTau/tau expression (Figure 5f).

### 2.5. Effect of Melatonin on the Production of Amyloid Beta in the SH-SY5Y Cells under OGD/R Conditions

Aβ production is one of the most important signaling events in AD. Based on the results on BACE1 and PS1 protein expression, we expected an increase in the production of Aβ under OGD/R conditions. Therefore, we used immunocytochemistry to detect the expression of Aβ and APP-C99. The results showed that OGD and OGD/R induced a high Aβ intensity in the SH-SY5Y cells. Compared with that in the cells subjected to OGD/R without melatonin, the intensity of Aβ in the cells subjected to OGD/R without melatonin decreased (Figure 6a,b). In the amyloidogenic pathway, BACE1 cleaves APP to generate transmembrane C99, which is subsequently processed by γ-secretase (including PS1) to generate the amyloid precursor protein (AICD), Aβ40, and Aβ42 in the cytoplasm [42]. Our immunostaining results showed that C99 was highly localized in the cytoplasm under OGD conditions, whereas the immunointensity in the OGD/R group treated with or without melatonin decreased (Figure 6c,d). Furthermore, Western blot analysis confirmed the findings from the immunostaining analysis and demonstrated a significant reduction in the C99 level in the OGD/R group treated with 50 and 100 µM of melatonin (Figure 6e,f) Aβ1-42 and Aβ1-40 are markers of amyloid pathology. Then, we examined whether OGD increased the levels of Aβ40 and Aβ42 by using multiplex ELISA. OGD significantly (*p* < 0.05) increased the levels of Aβ42 and Aβ40 and the ratio of Aβ42/40 (Figure 7a–c) in the SH-SY5Y cells. Treatment with 10 µM of melatonin during reoxygenation (OGD/R) significantly (*p* < 0.05) reduced the levels of Aβ42 and Aβ40 and the ratio of Aβ42/40 compared to those in the OGD/R group without melatonin (Figure 7a–c).

## 3. Discussion

Brain ischemia and AD are the main causes of irreversible disability and dementia worldwide [43,44,45,46,47,48,49]. There is no causal treatment that can stop the development of dementia in patients after both stroke and AD. It has been suggested that AD is a risk factor for stroke [10,11] and vice versa [12], indicating that the same or closely related pathological mechanisms may be involved in the development of both disorders. The pathophysiology of AD depends on oxidative stress, which is caused by Aβ aggregation and tau phosphorylation. Additionally, studies have shown that hypoxia may increase the production of Aβ, which may consequently contribute to the development of AD [50,51]. The main goal of this research was to investigate whether ischemia can induce AD pathogenesis and whether melatonin can attenuate these effects on the amyloidogenic pathway postischemia.

The OGD/R model was used to replicate the processes observed in human cerebral ischemia/reperfusion (I/R) injury. The ischemia and reperfusion model used in our study adhered to the Ryou and Mallet protocol for stroke treatment [52]. Here, we detected that insufficient oxygen and glucose altered the morphology and viability of the SH-SY5Y cells. We also validated the impact of OGD/R on hypoxia and the amyloidogenic pathway. We detected an increase in HIF-1α, BACE1, and PS1. Multiple studies have shown that the BACE1 gene promoter contains a biologically active region that binds to HIF-1α, and this binding can regulate the levels of BACE1 mRNA and protein [25,26,27,28]. HIF-1 is the main transcription factor involved in hypoxia. HIF-1 is a heterodimeric protein composed of an oxygen-regulated functional subunit, the HIF-1α subunit, and a complex with the HIF-1β subunit. HIF-1α is an oxygen-sensitive subunit, and it is expressed under hypoxic conditions. The HIF-1β subunit is consistently produced within the cells under normal oxygen conditions, whereas the HIF-1α subunit is degraded by oxygen-dependent HIF prolyl hydroxylase (PHD) through ubiquitin-dependent proteasomes. Under the conditions of low oxygen levels, the enzyme PHD becomes deactivated, thereby facilitating the stabilization and translocation of HIF-1α to the nucleus. However, the genes affected by HIF-1 play different roles across different diseases; for example, HIF-1α can induce transcriptional activation of oncogenic growth factor [53] and promote tumor metastasis [54,55], angiogenesis [56], vascular remodeling [57], glucose metabolism [58], and inflammation [59,60]. Furthermore, in ischemic cardiovascular disease, HIF-1 increases adenosine in the heart for cardioprotection during prolonged ischemia–reperfusion. HIF-1α plays a vital role in the development of cerebral ischemia by participating in numerous processes, including metabolism, proliferation, and angiogenesis [61].

BACE1 is a crucial enzyme in Aβ synthesis, and its mRNA and protein levels increase after ischemic stroke [62]. To investigate the effects of hypoxia on the BACE1 promoter, we used a reporter assay. The HIF-1α protein levels decreased in the OGD/R group compared to the OGD group. This decrease can be attributed to the rapid degradation of HIF-1α under normoxic conditions. Nevertheless, under the conditions of a low oxygen concentration (OGD), HIF-1α undergoes stabilization, resulting in elevated levels in hypoxic environments. Afterward, HIF-1α translocates to the nucleus and binds to its target genes, promoting transcription. The upregulation of HIF-1α resulted in an activation of the BACE1 promoter during OGD/R conditions, despite the decrease in the HIF-1α levels compared to those under the OGD conditions. This indicates that the regulation of BACE1 levels is influenced not only by HIF-1α but also by other factors. Several sources of evidence indicate that the BACE1 promoter contains multiple transcription factor-binding sites and exhibits inducible expression, including specificity protein 1 (SP1), nuclear factor kappa-light-chain-enhancer of activated B cells (NF-κB), and peroxisome proliferator-activated receptor gamma (PPARγ), which are involved in the cellular stress response [63,64,65].

Our results showed that reoxygenation combined with melatonin treatment decreased the protein expression and activity of the BACE1 promoter in the SH-SY5Y cells. The discovery of hypoxic response elements in the promoter of BACE1, where the transcription factor HIF-1α can bind during hypoxia, suggested that HIF-1α is the key mediator of the hypoxic response in BACE1. Several studies have shown that the promoters of BACE1 genes possess a biologically active HIF-1α binding region, and the binding of HIF-1α to the BACE1 promoter can regulate the levels of BACE1 mRNA and protein [25,26,27,28]. Consistent with our previous studies [35], melatonin decreased the BACE1 promoter activity under normoxic conditions.

Next, we examined the intracellular Aβ levels using multiplex ELISA and found that OGD/R combined with 10 µM of melatonin significantly reduced the hypoxia-induced excessive release of Aβ40 and Aβ42. Our data indicated that the increase in Aβ production due to hypoxia significantly increased the expression of BACE1 and PS1. Previous work has shown that inhibiting BACE1 leads to a significant decrease in cerebral Aβ levels, and a lack of BACE1 improved memory impairments in a mouse model of AD without causing significant adverse effects [66,67].

The tau hypothesis has also been proposed to explain the initial pathological trigger of AD. The excessive or abnormal phosphorylation of tau results in its transformation into paired helical filaments and neurofibrillary tangles, leading to neuronal death and dementia. Additional findings from our current study demonstrated that OGD/R might constitute an AD risk factor. In addition to increasing Aβ levels, OGD/R exposure modulates tau phosphorylation. Wen and colleagues described the molecular interactions that connect tau phosphorylation caused by ischemia to the development of AD [68]. Reoxygenation is essential for preventing organ damage after ischemia, while it can also generate reactive oxygen species (ROS). Previous reports have shown that melatonin treatment reduces ROS formation from amyloid beta in neuronal cells and improves cell survival [31]. Based on these findings, we might conclude that melatonin can be administered during reoxygenation to improve cell survival. Moreover, melatonin has the ability to prevent neuronal cell death resulting from tau phosphorylation in the brain after ischemia reperfusion.

In order to explore the interaction of melatonin with other signaling pathways involved in AD pathogenesis and provide a more comprehensive understanding of its therapeutic potential, Uzun’s group [69] conducted research on its interaction with other signaling pathways involved in AD pathogenesis. Their work presented a compelling case for the application of Intermittent Hypoxia–Hyperoxia Therapy (IHHT). Their systematic review delineates the therapeutic benefits of IHHT across a spectrum of pathologies, including but not limited to cardiovascular, respiratory, and metabolic syndromes, which are often comorbid with AD. The mechanistic basis of these benefits, attributed to the modulation of hypoxia-inducible factors and the enhancement of cellular resilience to fluctuating oxygen levels, aligns with our findings on the neuroprotective effects of melatonin in hypoxic environments. Therefore, integrating IHHT could potentially amplify the therapeutic efficacy of melatonin by leveraging these complementary mechanisms of action, suggesting a promising avenue for future research in AD treatment strategies.

## 4. Materials and Methods

### 4.1. Cell Culture and Treatments under Oxygen–Glucose Deprivation (OGD) and Reoxygenation (OGD/R) Conditions

The human dopaminergic neuroblastoma SH-SY5Y cell lines were obtained from the American Type Culture Collection (Manassas, VA, USA). The SH-SY5Y cells were cultured in modified Eagle’s medium supplemented with a nutrient mixture of Ham F12 (MEM/F12) supplemented with 10% heat-inactivated fetal bovine serum (FBS) (Gibco–Thermo Fisher Scientific, Waltham, MA, USA), 100 U/mL penicillin, and 100 µg/mL streptomycin. The cells were cultured at 37 °C in 5% CO_2_ and 95% air for 18–24 h before exposure to hypoxia. The medium was changed to DMEM without glucose, pyruvate, or FBS before the cells were incubated in a humidified hypoxic chamber with 1% O_2_, 5% CO_2_, and 95% N_2_ at 37 °C for 4 h. Following OGD exposure, the medium was replaced with conditioned medium, and the cultured cells were then returned to a 37 °C CO_2_ incubator under reoxygenation conditions for 24 h. The cells were collected after 4 h of OGD exposure and OGD/R treatment with or without 10 µM or 100 µM of melatonin [70].

### 4.2. Trypan Blue Assay

The effects of melatonin on the viability of the OGD/R-treated SH-SY5Y cells were examined. After oxygen–glucose deprivation and reoxygenation, the cells were trypsinized for 3 min at 37 °C and centrifuged for 5 min at 1500 rpm. The supernatant was removed, and the cells were resuspended in 1× PBS. The cells were then mixed with 0.4% trypan blue dye at a ratio of 1:1, and trypan-blue-positive cells were observed using light microscopy.

### 4.3. Cell Viability Assay

Cell viability was determined using a cell count and viability kit (Cat. No MCH100102) according to the manufacturer’s instructions. Briefly, the cells were trypsinized for 3 min at 37 °C and centrifuged for 5 min at 1500 rpm. The cells were then incubated with Muse Count and Viability Reagent for 5 min at room temperature in the dark. The stained cells were immediately analyzed using a Guava Muse cell analyzer (Merck-Millipore, Temecula, CA, USA).

### 4.4. Quantitative RT-PCR

Total RNA from the OGD/R-induced cells was extracted using a GF1 RNA kit (Vivantis). Briefly, 1 µg of purified RNA from each sample was converted into cDNA using reverse transcription polymerase chain reaction (SensiFAST cDNA Synthesis Kit) (Meridian Bioscience, Memphis, TN, USA). The cDNAs of the HIF-1α, APP, TAU, BACE1, and PS1 genes and the internal control GAPDH were amplified using specific primers, as shown in Table 1. The PCRs were performed using SYBR^®^ Green PCR Master Mix (SensiFAST SYBR NO-ROX kit) (Meridian Bioscience, Memphis, TN, USA). The PCRs were performed as follows: denaturation at 95 °C for 2 min, followed by 40 cycles of 10 s at 95 °C, combined with annealing at 60 °C for 5 s, according to the manufacturer’s instructions.

### 4.5. Western Blot Analysis

After OGD and OGD/R, the cells were harvested, washed, and lysed with RIPA lysis buffer (50 mM of Tris base, 150 mM of NaCl, 1 mM of EDTA, 1 mM of phenyl methane sulfonyl fluoride (PMSF), 1% Triton X-100, 1% protease inhibitor, 1% phosphatase inhibitor, and 0.1% SDS). The cell lysate was sonicated and subsequently centrifuged at 12,000× *g* at 4 °C for 15 min. The total protein concentration was quantified using Bradford reagent (Himedia, Maharashtra, India). After the addition of sample loading buffer, 20 µg of the protein sample was subjected to 12% SDS-PAGE. A constant voltage of 120 V was applied, and the device was coupled with a Bio-Rad apparatus. The protein samples were transferred to PVDF membranes at 100 V for 2 h. Subsequently, the PVDF membranes were blocked at room temperature with 3% bovine serum albumin or 5% nonfat milk in TBST for 1 h and incubated at 4 °C overnight with primary antibodies, as shown in Table 2. The blots were then incubated with horseradish peroxidase (HRP)-conjugated secondary anti-rabbit or anti-mouse antibodies. The signals on the membrane were subsequently developed with a chemiluminescent ECL reagent (ThermoFisher Scientific, St. Louis, MO, USA).The signals were detected using an Amersham ImageQuant 800 (Cytiva, Marlborough, MA, USA) and analyzed using ImageQuant TL 1D Software version 8.2. The intensities of the immunoblot bands were normalized using β-actin as an internal standard.

### 4.6. Enzyme-Linked Immunosorbent Assay

The cell lysates were collected and sonicated in ice-cold lysis buffer. The lysate was centrifuged at 12,000× *g* for 10 min at 4 °C. The supernatant was collected to measure human Aβ1-40 and human Aβ1-42 (pg/mg protein) according to the manufacturer’s instructions by using a 4-plex custom MIILIPLEX kit (human Aβ40/42 ELISA kit) (Millipore, Merck, Darmstadt, Germany). The fluorescent signals were detected using MAGPIX^®^ Luminex’s xMAP^®^ instrument xPONENT 4.3. The concentrations of each marker were analyzed using Belysa^®^ Curve Fitting Software version 1.2. The ratio of Aβ42/40 was further calculated.

### 4.7. Immunocytochemistry

The SH-SY5Y cells were incubated for 24 h in a 24-well plate containing complete media and a coverslip precoated with poly-L-lysine to allow them to glow and attach to the coverslip. The cells were then incubated under OGD conditions with or without reoxygenation and reoxygenation with 100 µM of melatonin. After that, the cells attached to the coverslips were fixed with 4% paraformaldehyde in phosphate-buffered saline (1× PBS) for 10 min at room temperature, followed by washing with PBS three times before permeabilization in PBS supplemented with 0.1% Triton X-100 (PBS-T) for 10 min and blocking with 20% fetal bovine serum in PBS-T for 30 min at room temperature. After washing, the cells on the coverslips were incubated with primary rabbit anti-HIF-1alpha antibody (1:200), rabbit anti-Aβ antibody (1:200), and mouse anti-APP-C99 (1:200) overnight at 4 °C and then with Alexa 568-conjugated anti-rabbit and Alexa 488-conjugated anti-mouse secondary antibodies. After washing them in PBS, the nuclei were stained with 10 µM of Hoechst 33342 (Invitrogen, Carlsbad, CA, USA). Coverslips were mounted over the samples with fluorescent mounting media (Sigma, Riverside, CA, USA), and the stained proteins were visualized under a confocal laser scanning microscope (FV3000, Olympus, Tokyo, Japan).

### 4.8. Luciferase Assay

The pGL3-BACE1-luc plasmid, containing the human BACE1 promoter region upstream of the firefly luciferase reporter gene [26,63], was used for the luciferase assay to determine the activity of the BACE1 promoter. SH-SY5Y cells grown in 96-well plates were transfected with 0.1 µg of the promoter construct using Lipofectamine 2000 (Invitrogen, Carlsbad, CA, USA). At 48 h post-transfection, the cells were subjected to hypoxia for 4 h and reoxygenation with or without 100 µM of melatonin for 24 h. The luciferase assay was performed according to the protocol for the luciferase assay system (Gold Bio, St Louis, MO, USA), and the relative light intensity was measured using a luminometer (BioTek, San Diego, CA, USA) to determine the luciferase activity.

### 4.9. Statistical Analysis

For all the experiments, the data were analyzed, and statistical significance was determined using GraphPad Prism version 8.3.0. Differences between two groups were assessed using unpaired *t* tests (where equal variances were not assumed). Differences between multiple groups were assessed using a one-way analysis of variance (ANOVA), followed by Tukey’s multiple comparison test (where equal variances were not assumed). The data are expressed as the mean ± S.E.M. *p* values < 0.05 were considered to indicate statistical significance.

## 5. Conclusions

This study demonstrated that OGD/R upregulates the expression of BACE1, resulting in an increase in the ratio of amyloid-β production (both Aβ40 and Aβ42) and tau hyperphosphorylation. We are the first to demonstrate that HIF-1α is the key mediator of the hypoxic response in BACE1 and that reoxygenation with melatonin can decrease the nuclear translocation of HIF-1α and BACE1 expression (Figure 8). We may conclude that hypoxia regulates the amyloid pathway, while melatonin decreases hypoxia conditions and APP processing. This study is the first to demonstrate that melatonin has the potential to inhibit the development of Alzheimer’s disease pathogenesis in IS patients.

## Figures and Tables

**Figure 1 ijms-25-05225-f001:**
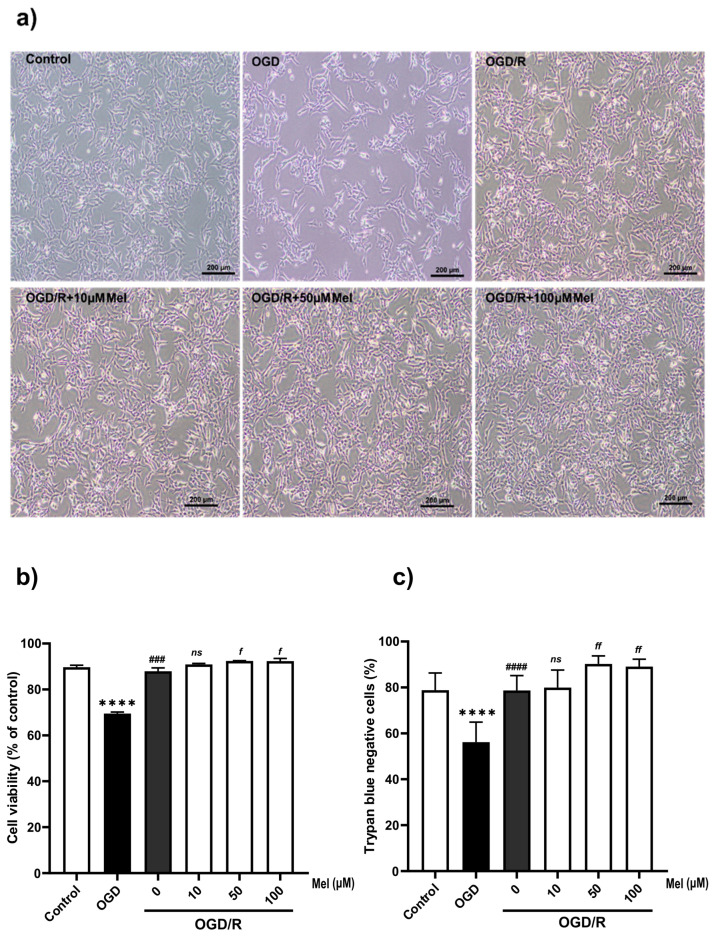
The effect of melatonin on OGD/R-induced morphological changes and cell death in SH-SY5Y cells. The cells were incubated with OGD for 4 h and then reoxygenated for 24 h with various concentrations of melatonin. (**a**) Representative bright-field microscopy (10×) images of cell morphology (control, OGD, OGD/R without or with 10, 50, and 100 µM of melatonin). (**b**) Cell viability was quantified using the Muse cell viability kit and (**c**) trypan blue assay. All experiments were performed in triplicate and repeated at least twice. The data are presented as the means ± S.E.M.s, and Student’s unpaired *t* test was performed. (**** denotes statistical significance at *p* < 0.0001 compared to the control group; ### and #### denote statistical significance at *p* < 0.001 and 0.0001 compared to the OGD group; and *f* and *ff* denote statistical significance at *p* < 0.05 and 0.01 compared to the OGD/R group; *ns*, not significant, *p* > 0.05, respectively).

**Figure 2 ijms-25-05225-f002:**
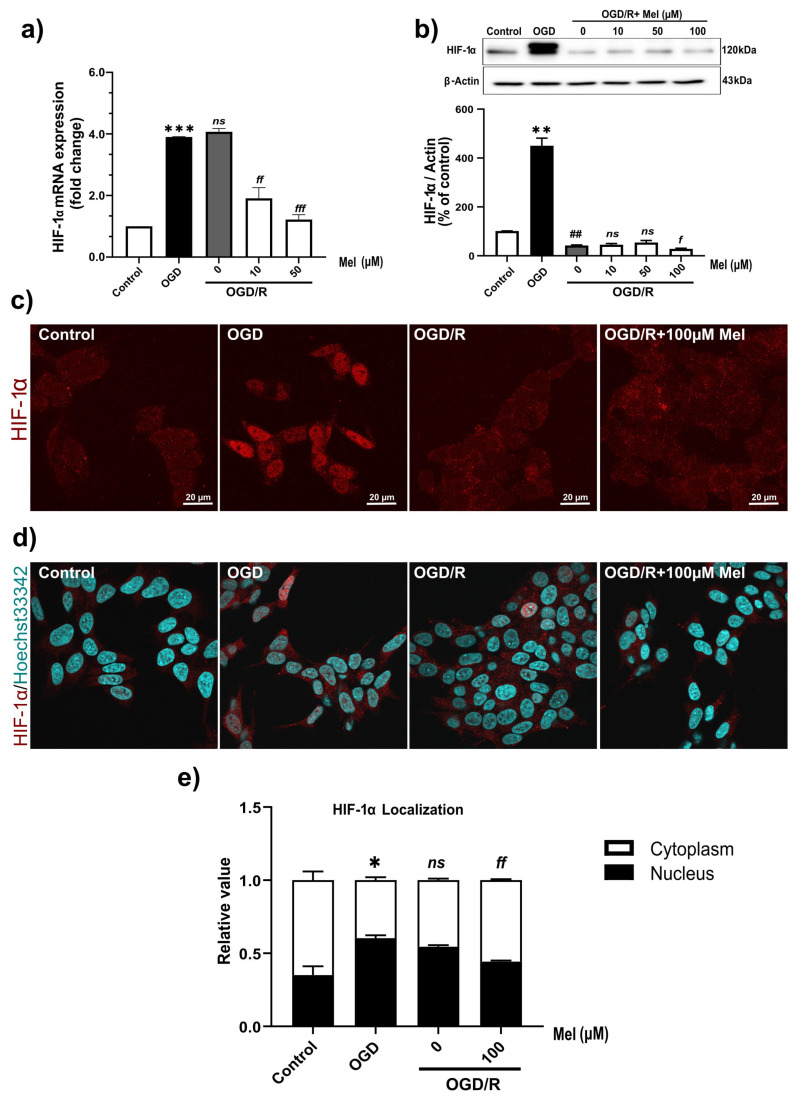
Effects of melatonin on the expression and localization of HIF-1α at *p* < 0.05 under OGD/R conditions. (**a**) Expression of HIF-1α was confirmed using qPCR. (**b**) Western blotting analysis. (**c**,**d**) Confocal microscopy analysis of HIF-1α localization in OGD/R-treated cells treated with or without melatonin; 100× magnification. Immunocytochemistry of SH-SY5Y cells revealed HIF-1α expression and localization in both the nucleus and cytoplasm. Cyan: nucleus; red: HIF-1α. (**e**) Nuclear/cytoplasmic localization of HIF-1α. All experiments were performed in triplicate and repeated at least twice. Quantitative analysis of the localization of HIF-1α in the nucleus and cytoplasm using ImageJ Software 1.54f (NIH Image, Bethesda, MD, USA). The data are presented as the means ± S.E.Ms. (*n* = 3), and Student’s unpaired *t* test was performed. *, **, and *** denote statistical significance at *p* < 0.05, 0.01, and 0.001 compared to the control group; ## denote statistical significance at *p* < 0.01 compared to the OGD group; and *f*, *ff*, and *fff* denote statistical significance at *p* < 0.05, 0.01, and 0.001 compared to the OGD/R group; *ns*, not significant, *p* > 0.05, respectively.

**Figure 3 ijms-25-05225-f003:**
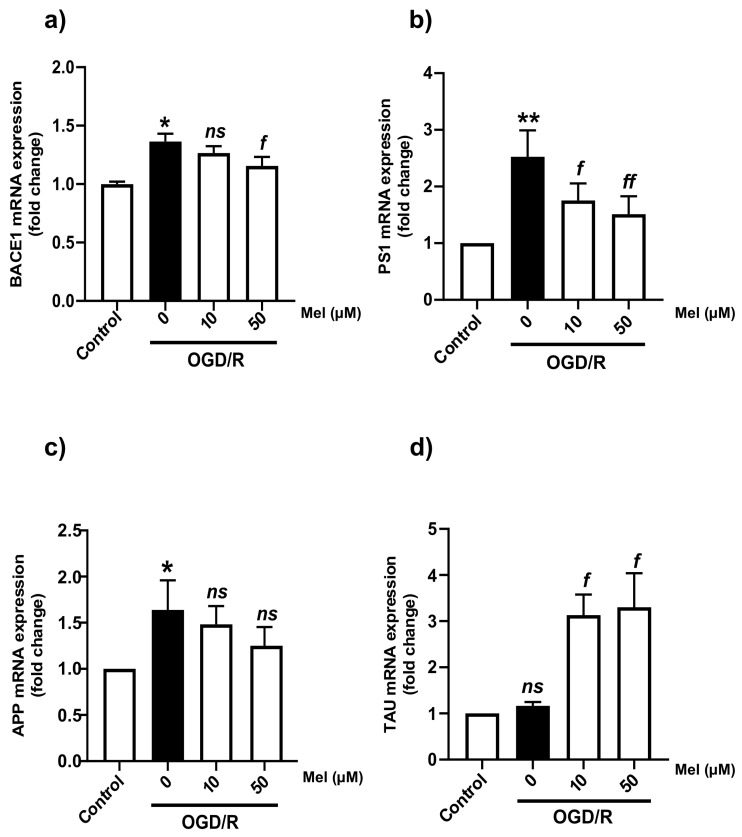
Effect of melatonin on the expression of mRNAs involved in the amyloid pathway in SH-SY5Y cells under OGD/R conditions. The cells were incubated with OGD for 4 h and then reoxygenation (OGD/R) for 24 h with 10 µM and 50 µM of melatonin. (**a**–**d**) Quantitative real-time PCR was used to determine the mRNA levels of BACE1, PS1, APP, and TAU in SH-SY5Y cells under OGD/R conditions. All experiments were performed in triplicate and repeated (*n* = 3). The data are presented as the means ± S.E.M.s, and Student’s unpaired *t* test was performed. * and ** denote statistical significance at *p* < 0.05 and 0.01, respectively, compared to the control group; *f* and *ff* denote statistical significance at *p* < 0.05 and 0.01, respectively, compared to the OGD/R group; *ns*, not significant, *p* > 0.05.

**Figure 4 ijms-25-05225-f004:**
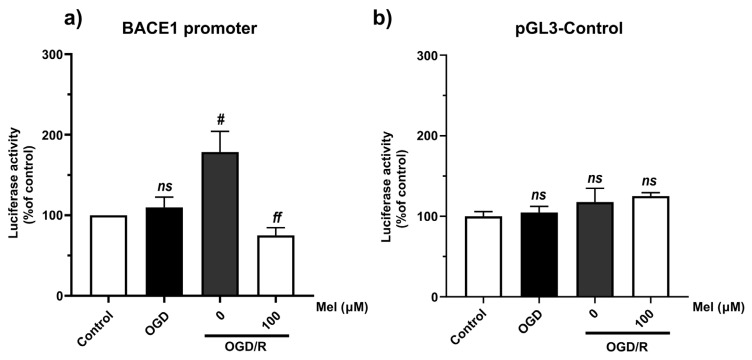
Effect of melatonin on increasing BACE1 promoter activity under oxygen–glucose deprivation and reoxygenation conditions. A luciferase assay was performed to evaluate the effects of BACE1 and APH1 promoter activation. SH-SY5Y cells were transfected with the BACE1 promoter–luciferase reporter construct (pGL3-BACE1-luc). The transfected cells were then exposed to OGD/R with or without 100 µM of melatonin for 24 h. (**a**) BACE1 promoter activity. (**b**) Luciferase activity was normalized to that of the pGL3 control in each sample. The data are presented as the means ± S.E.Ms. (*n* = 3) (# denotes statistical significance at *p* < 0.05 compared to the OGD group; *ff* denotes statistical significance at *p* < 0.01 compared to the OGD/R group; *ns*, not significant, *p* > 0.05).

**Figure 5 ijms-25-05225-f005:**
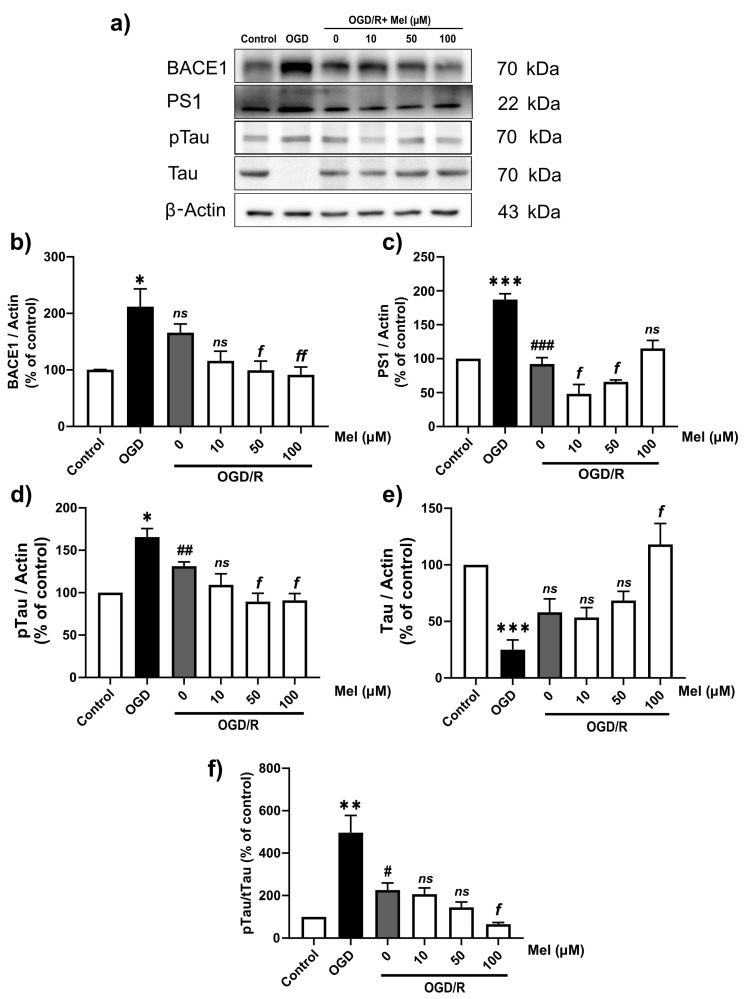
Effect of melatonin on the protein levels of components of the amyloidogenic pathway in SH-SY5Y cells under OGD/R conditions. The cells were incubated with OGD for 4 h and then reoxygenated for 24 h with 10, 50, or 100 µM of melatonin. (**a**) Western blotting was used to determine the protein expression of BACE1, PS1, tau, and pTau in SH-SY5Y cells under OGD/R conditions. The band densities were normalized to those of actin. (**b**) BACE1, (**c**) PS1, (**d**) pTau, (**e**) Tau, (**f**) pTau/Tau. The ratios were calculated as a percentage of the respective value of the control group. The data represent the means ± S.E.Ms. (*n* = 3–5) (*, **, and *** denote statistical significance at *p* < 0.05, 0.01, and 0.001 compared to the control group, respectively; #, ##, and ### denote statistical significance at *p* < 0.05, 0.01, and 0.001 compared to the OGD group; *f* and *ff* denote statistical significance at *p* < 0.05 and 0.01, respectively, compared to the OGD/R group; *ns*, not significant, *p* > 0.05.

**Figure 6 ijms-25-05225-f006:**
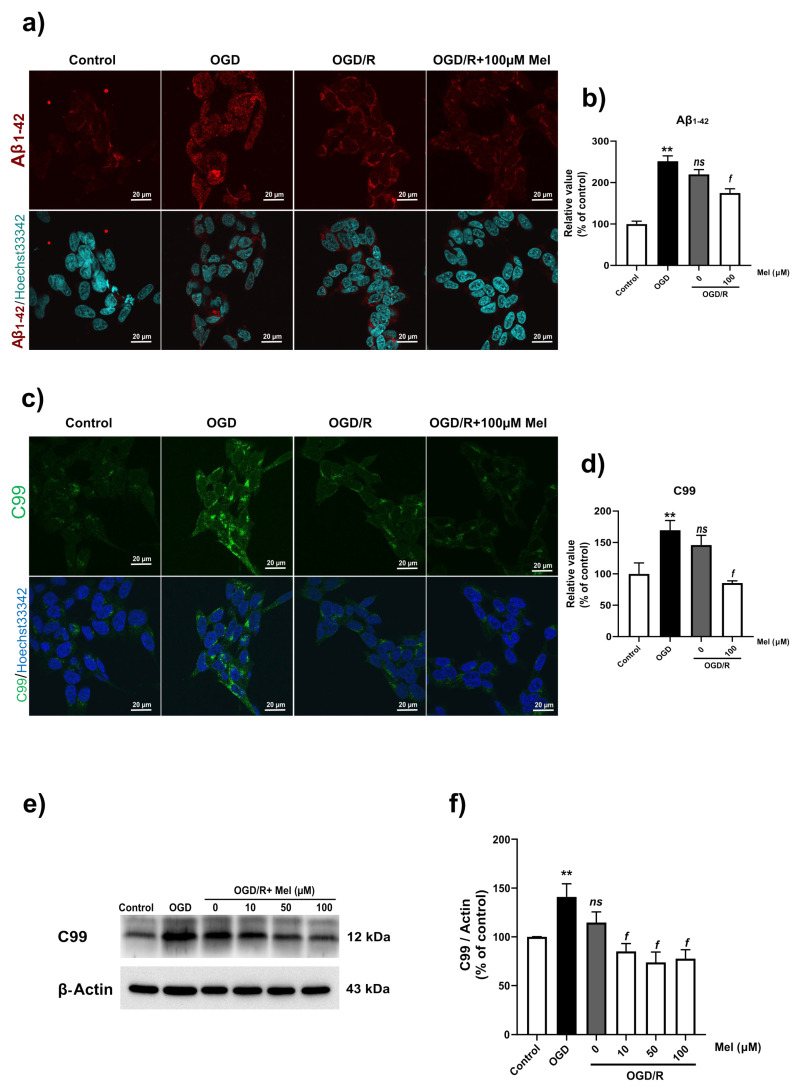
Expression of amyloid protein in SH-SY5Y cells under OGD/R conditions. The cells were incubated with OGD for 4 h and then reoxygenated for 24 h with 100 µM of melatonin. (**a**,**b**) Immunostaining was used to determine the protein levels of Aβ. (**c**,**d**) Levels of APP-C99 in SH-SY5Y cells under ODG/R conditions, after which the cells were observed under a confocal microscope (100×). (**e**,**f**) Western blotting was used to determine the protein expression of APP-C99 in SH-SY5Y cells under OGD/R conditions. The band densities were normalized to those of actin. The data represent the means ± S.E.Ms. (*n* = 3) (** denotes statistical significance at *p* < 0.01 compared to the control group, and *f* denotes statistical significance at *p* < 0.05 compared to the OGD/R group; *ns*, not significant, *p* > 0.05, respectively).

**Figure 7 ijms-25-05225-f007:**
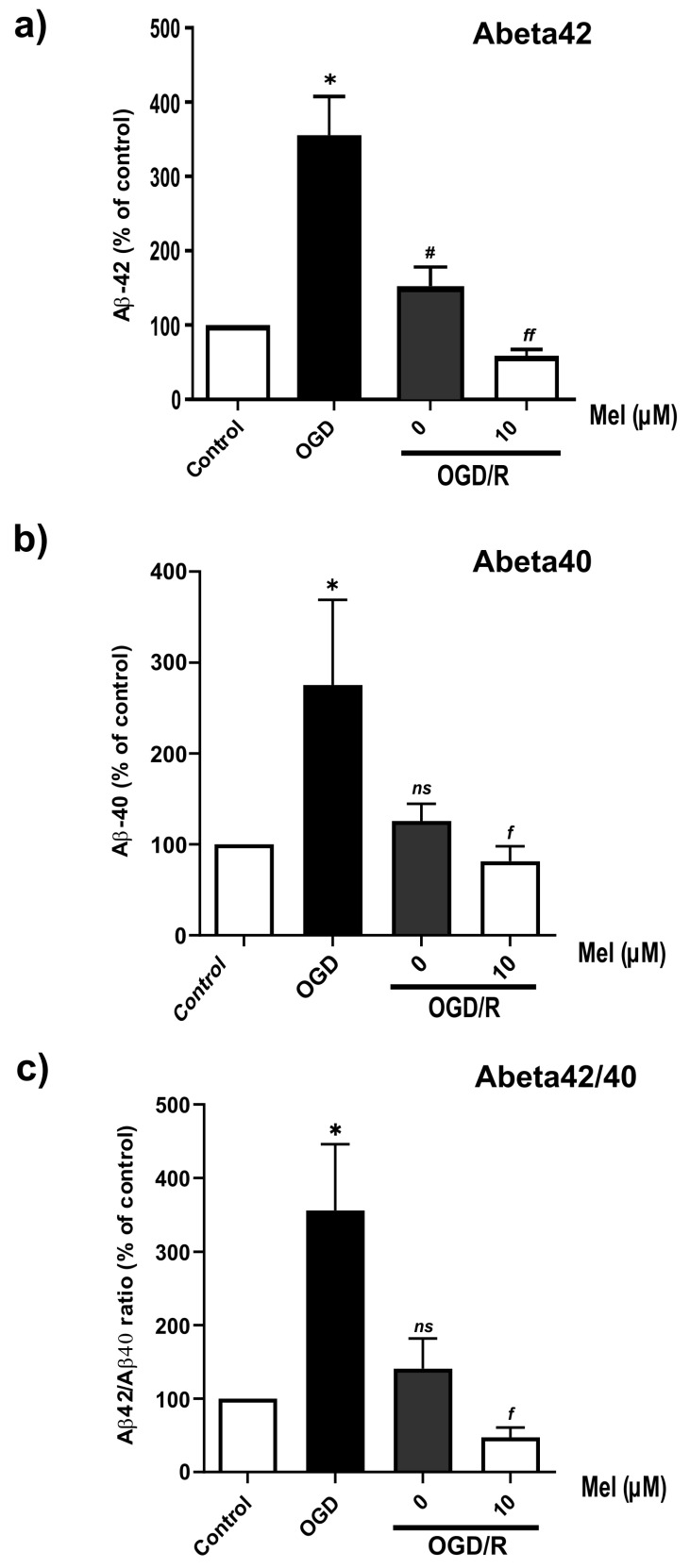
Effect of melatonin on the Aβ42/40 ratio in SH-SY5Y cells under OGD/R conditions. The cells were incubated with OGD for 4 h and then reoxygenated (OGD/R) for 24 h with 10 µM of melatonin. The protein lysates were harvested and analyzed using multiplex ELISA for human Aβ1-40 and Aβ1-42. (**a**–**c**) Ratios were calculated as a percentage of the respective value of the control group. The data are presented as the means ± S.E.Ms. (*n* = 3) (* denotes statistical significance at *p* < 0.05 compared to the control group, and # denotes statistical significance at *p* < 0.05 compared to the OGD group; *f* and *ff* denote statistical significance at *p* < 0.05 and 0.01 compared to the OGD/R group, respectively; *ns*, not significant, *p* > 0.05).

**Figure 8 ijms-25-05225-f008:**
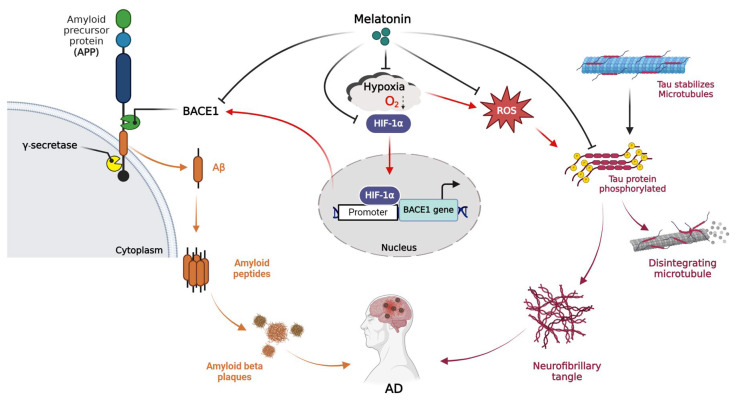
Hypoxia is a common risk factor for both Alzheimer’s disease (AD) and ischemic stroke. Hypoxia-inducible factor-1α (HIF-1α), an indicator of ischemic stroke, is highly upregulated during oxygen–glucose deprivation. It is a transcriptional factor responsible for cellular and tissue adaptation to low oxygen tension. HIF-1α upregulated the β-site APP-cleaving enzyme 1 (BACE1) by activating the BACE1 promoter, resulted in the generation of amyloid beta (Aβ). In addition, it promotes tau hyperphosphorylation, which further exacerbates AD. The present results demonstrate that HIF-1α is the key mediator of the hypoxic response in BACE1 and that reoxygenation with melatonin can decrease the nuclear translocation of HIF-1α and BACE1. We may conclude that hypoxia upregulates the amyloid pathway, while melatonin decreases hypoxia conditions and APP processing. This suggests that melatonin has the potential to inhibit the development of Alzheimer’s disease pathogenesis in ischemic stroke patients. This graphic was created using BioRender (https://biorender.com/, accessed on 11 March 2024).

**Table 1 ijms-25-05225-t001:** The sequences of primers used for real-time PCR.

Gene Name	GenBank	Forward (5’3’)	Reverse (5’3’)
APP	NM_001136131.3	GCTGGCCTGCTGGCTGAACC	GCGACGGTGTGCCAGTGAA
TAU	XM_054316146.1	GACAGAGTCCAGTCGAAGATTG	AGGAGACATTGCTGAGATGC
BACE1	NM_001411039.1	AGGTTACCTTGGCGTGTGTC	GAGGCAATCTTTGCACCAAT
PS1	XM_054376420.1	AATAGAGAACGGCAGGAGCA	GCCATGAGGGCACTAATCAT
HIF1α	NM_001243084.2	CTTGCTCATCAGTTGCCACTTC	GCCATTTCTGTGTGTAAGCATTTC
GAPDH	NM_001357943.2	ACAACTTTGGTATCGTGGAAGG	GCCATCACGCCACAGTTTC

**Table 2 ijms-25-05225-t002:** The list of antibodies.

Antibodies	Catalogue Number	Dilution	Source
Rabbit anti-Aβ1-42	14947	1:2000	Cell Signaling Technology, Inc., Danvers, MA, USA
Mouse anti-APP-C99	MABN380	1:2000	Merck Millipore, Temecula, CA, USA
Rabbit anti-PS1	3622S	1:1000	Cell Signaling Technology, Inc., Danvers, MA, USA
Rabbit anti-HIF1α	Ab179483	1:1000	Abcam, Cambridge, UK
Rabbit anti-BACE1	Ab108394	1:2000	Abcam, Cambridge, UK
Mouse anti-pTau (Thr181)	MABN388	1:500	Merck Millipore, Temecula, CA, USA
Mouse anti-Tau	05348	1:1000	Merck Millipore, Temecula, CA, USA
Mouse anti-β-Actin	MAB1501	1:20,000	Merck Millipore, Temecula, CA, USA
Goat anti-Mouse IgG	AP142P	1:1000–1:20,000	Merck Millipore, Temecula, CA, USA
Goat anti-Rabbit IgG	AP132P	1:1000–1:20,000	Merck Millipore, Temecula, CA, USA

## Data Availability

The data are contained within the article.

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
