# Peer review of "Melatonin Inhibits Hypoxia-Induced Alzheimer’s Disease Pathogenesis by Regulating the Amyloidogenic Pathway in Human Neuroblastoma Cells"

_ijms, 2024, doi:10.3390/ijms25105225_

Round 1

Reviewer 1 Report

Comments and Suggestions for Authors

This study by Singrang et al explores the potential of melatonin in mitigating the effects of ischemic stroke that can potentially contribute to Alzheimer's disease (AD). There are several concerns that need to be addressed before the manuscript is acceptable for publication:

1.  The introduction lacks the big picture and novelty of the research. 

2. Figure 1. Is there a way to quantify the change in morphology of the cells? For fig 1b, the results are not very convincing. 50uM Melatonin and seems to have the same effect as 10uM. To me it does not look statistically significant.

3. Figure 2b. HIF-1a seems to have two bands, why can't we see the top band for the  conditions other than OGD? Also, which band has been quantified for OGD?Perhaps it will be better to show individual data points for the western blot graphs (from all the experiments).

4. For sections 2.3 and 2.4, the authors conclude:

Lines (170-171): Our results indicated that hypoxia regulated BACE1 170 transcription, whereas melatonin decreased hypoxia conditions and APP processing. 

Lines (199-201)Our results indicated that hypoxia regulated BACE1, which is the major β-secretase enzyme complex for the production of amyloid-β peptide, while melatonin decreased APP processing.

These conclusions are overreaching especially because for these sections the authors have not yet looked at Abeta fragments. It makes more sense to make these conclusions in the latter sections.

5. Figure 5a Label the lanes for the western blot. The results for PS1 and Tau do not look convincing. It appears that the low intensity bands are because of improper transfer of the proteins to the membrane. The authors need to provide rationale in the text for why they are looking at PS1 and pTau levels.

6. Figure 6. It will be informative to quantify the intensity of Abeta1-42 and C99 levels for the immunostaining. From the western blot results (fig 6c), it appears that 10uM Melatonin has the same effect as 100uM. Why did the authors choose 100uM for immunostaining instead of 10uM?

7. In the Discussion section, the authors mention "HIF-1α protein levels decreased in the OGD/R group compared to the OGD group."  The authors need to discuss the implications of this result in detail.

Comments on the Quality of English Language

Minor editing of English is required. Some sentences do not make sense.For example: 

Line 35-36: Stroke is classified as intracerebral hemorrhage and ischemic stroke (IS), leading to a disruption in the essential blood flow of the brain. IS occurs due to thrombotic obstruction

Author Response

Reviewer 1.

Reviewer 2 Report

Comments and Suggestions for Authors

1. I advise exploring the interaction of melatonin with other signaling pathways involved in AD pathogenesis to provide a more comprehensive understanding of its therapeutic potential.

2. In exploring potential adjunct therapies to melatonin for combating hypoxia-induced pathogenesis in Alzheimer's Disease (AD), the work of Uzun et al. (2023) presents a compelling case for the incorporation of Intermittent Hypoxia–Hyperoxia Therapy (IHHT). Their systematic review delineates the therapeutic benefits of IHHT across a spectrum of pathologies, including but not limited to cardiovascular, respiratory, and metabolic syndromes, which are often comorbid with AD. The mechanistic basis of these benefits, attributed to the modulation of hypoxia-inducible factors and enhancement of cellular resilience to fluctuating oxygen levels, aligns with your findings on the neuroprotective effects of melatonin in hypoxic environments. Therefore, integrating IHHT could potentially amplify the therapeutic efficacy of melatonin by leveraging these complementary mechanisms of action, suggesting a promising avenue for future research in AD treatment strategies. Uzun, A.-B.; Iliescu, M.G.; Stanciu, L.-E.; Ionescu, E.-V.; Ungur, R.A.; Ciortea, V.M.; Irsay, L.; MotoaÈ™că, I.; Popescu, M.N.; Popa, F.L.; et al. Effectiveness of Intermittent Hypoxia–Hyperoxia Therapy in Different Pathologies with Possible Metabolic Implications. Metabolites 202313, 181. https://doi.org/10.3390/metabo13020181

Comments on the Quality of English Language

In some instances, sentences could be simplified to improve understanding without sacrificing scientific accuracy.

Author Response

Reviewer 2.

Reviewer 3 Report

Comments and Suggestions for Authors

The manuscript with the ID 2951007, submitted to the International Journal of Molecular Sciences, explores the impact of melatonin on hypoxia-induced Alzheimer’s disease (AD) pathogenesis by regulating the amyloidogenic pathway in human neuroblastoma cells. Authored by N. Singrang, the paper investigates melatonin's potential to prevent the aberrant development of ischemic stroke, a precursor to AD. The study aims to evaluate whether oxygen-glucose deprivation/reoxygenation (OGD/R) conditions can upregulate the amyloidogenic pathway's activity and whether melatonin pretreatment can ameliorate hypoxia's effects in SH-SY5Y cells.

The authors further explore melatonin's influence on cell survival and its relationship with hypoxia and amyloidogenic processing, correlating these effects with the hypoxia marker HIF-1α. Neuroblastoma cells (SH-SY5Y) subjected to OGD and OGD/R conditions serve as a model to address the study's objectives. Utilizing cell viability assays, qRT-PCR, ELISA, immunohistochemistry, and luciferase assays, the authors present findings that support the hypotheses outlined in the manuscript.

My comments are as follows:

 Abstract

This section is well-written, presenting the background and results concisely and convincingly.

Introduction

This section adequately supports the background and purpose of the study.

M&M

It is evident that the authors utilized undifferentiated SH-SY5Y cells. My query arises as to why the cells were not differentiated to better resemble the real system rather than using solely neuroblastoma cells. I am curious if such experiments were conducted.

The focus on testing OGD/R and the effect of melatonin in hypoxic conditions is particularly intriguing. Additionally, I am interested in whether melatonin has been shown to modulate various signaling pathways involved in cell survival and death.

The manuscript mentions that the concentration of melatonin ranged between 10-100uM. I am curious if lower concentrations of melatonin were tested. What is the physiological concentration of melatonin, and what effect does lower concentration have on cell survival after OGD/R condition? How does melatonin affect neuronal cell death/survival at such concentrations?

In Figure 6, an increased level of Ab42 is shown. I wonder about the expression level of APP in these cells. Furthermore, in Figure 6a, the blot labeling is missing (e.g., control, OGD, etc.).

Regarding the decreased level of APP mRNA in OGD/R after melatonin treatment, what is the APP mRNA level at OGD? If the APP mRNA level decreases after melatonin treatment under OGD/R conditions, then the possibility of APP cleavage by γ-secretase should be reduced, leading to a decrease in Abeta 42.

Overall, the outcome of these findings suggest that melatonin may have therapeutic potential in mitigating neuronal damage associated with ischemic conditions like stroke. However, further studies are needed to elucidate the precise mechanisms underlying melatonin's neuroprotective effects in the context of OGD/R injury in SH-SY5Y cells.

Author Response

Reviewer 3.

Round 2

Reviewer 1 Report

Comments and Suggestions for Authors

There are still some concerns that I have:

For Figure 6, I had suggested that it will be informative to quantify the intensity of Abeta1-42 and C99 levels for the immunostaining. The authors have not responded to this suggestion.

Comments on the Quality of English Language

There are still several grammatically incorrect sentences. Some examples include:

A. According to reported that HIF-1α, a central actor in adaptive response to hypoxia, showed a progressive increase in HIF-1α phosphorylated forms during the hypoxia periods.

B. In order to explore the interaction of melatonin with other signaling pathways involved in AD pathogenesis to provide a more comprehensive understanding of its therapeutic potential, the work of Uzun’s group presented a compelling case for the incorporation of Intermittent Hypoxia–Hyperoxia Therapy (IHHT) has been included as the following. 

Author Response

Suggestions for Authors

For Figure 6, I had suggested that it will be informative to quantify the intensity of Abeta1-42 and C99 levels for the immunostaining. The authors have not responded to this suggestion.

ANS: The intensity of Abeta -42 and C99 levels for the immunostaining have been quantified in Figure 6b and Figure 6d, respectively.   

Comments on the Quality of English Language

There are still several grammatically incorrect sentences. Some examples include:

  1. According to reported that HIF-1α, a central actor in adaptive response to hypoxia, showed a progressive increase in HIF-1α phosphorylated forms during the hypoxia periods.

ANS: Previous finding from Toffoli’s group (39) reported that HIF-1α exhibited a progressive increase in phosphorylated forms throughout the course of the hypoxic period.

  1. In order to explore the interaction of melatonin with other signaling pathways involved in AD pathogenesis to provide a more comprehensive understanding of its therapeutic potential, the work of Uzun’s group presented a compelling case for the incorporation of Intermittent Hypoxia–Hyperoxia Therapy (IHHT) has been included as the following.

ANS: In order to explore the interaction of melatonin with other signaling pathways

involved in AD pathogenesis and provide a more comprehensive understanding of its therapeutic potential, Uzun’s group [69] conducted research on its interaction with other signaling pathways involved in AD pathogenesis. Their work presented a compelling case for the application of Intermittent Hypoxia-Hyperoxia Therapy (IHHT).  
